# A Smartphone App for Improving Mental Health through Connecting with Urban Nature

**DOI:** 10.3390/ijerph16183373

**Published:** 2019-09-12

**Authors:** Kirsten McEwan, Miles Richardson, David Sheffield, Fiona J. Ferguson, Paul Brindley

**Affiliations:** 1Human Sciences Research Centre, The University of Derby, Derby DE22 1GB, UK; K.McEwan@derby.ac.uk (K.M.); D.Sheffield@derby.ac.uk (D.S.); fiona-j-ferguson@hotmail.co.uk (F.J.F.); 2Department of Landscape Architecture, The University of Sheffield, Sheffield S10 2TN, UK; P.Brindley@sheffield.ac.uk

**Keywords:** mental health, wellbeing, green space, mobile app, nature connectedness, social prescription, urban

## Abstract

In an increasingly urbanised world where mental health is currently in crisis, interventions to increase human engagement and connection with the natural environment are one of the fastest growing, most widely accessible, and cost-effective ways of improving human wellbeing. This study aimed to provide an evaluation of a smartphone app-based wellbeing intervention. In a randomised controlled trial study design, the app prompted 582 adults, including a subgroup of adults classified by baseline scores on the Recovering Quality of Life scale as having a common mental health problem (*n* = 148), to notice the good things about urban nature (intervention condition) or built spaces (active control). There were statistically significant and sustained improvements in wellbeing at one-month follow-up. Importantly, in the noticing urban nature condition, compared to a built space control, improvements in quality of life reached statistical significance for all adults and clinical significance for those classified as having a mental health difficulty. This improvement in wellbeing was partly explained by significant increases in nature connectedness and positive affect. This study provides the first controlled experimental evidence that noticing the good things about urban nature has strong clinical potential as a wellbeing intervention and social prescription.

## 1. Introduction

Mental illness is the largest cause of disability in the United Kingdom (UK), contributing to 22.8% of the total burden of disease [1]. The wider economic cost of mental illness is estimated at £105.2 billion per year in the UK [2] and 30% of the global population has suffered from a mental disorder [3]. It is increasingly accepted that exposure to the natural environment is linked to human health and wellbeing (for reviews, see [4,5,6]). Interventions to increase human engagement and connection with the natural environment are widely-accessible and cost-effective ways of improving human wellbeing and reducing health inequalities [7]. The importance of having access to nearby or urban green space is recognised in policy, with the European Environment Agency recommending that people should have access to green space within a 15-min walk from their home, the UK Government developing a 25-year plan to increase the connection between people and nature [8], and the World Health Organisation stating that urban green space is a “necessary component for delivering healthy, sustainable, liveable conditions” [1]. However, with increased urbanisation [9] there are fewer opportunities for people to access and engage with nature.

Urban natural environments provide daily access to residents who would not normally have the time or inclination to travel further distances to natural environments [10]. Therefore, interventions are needed to connect people with urban nature close to home [11,12]. Indeed, close to home urban natural environments providing day-to-day stress relieving effects have been seen as crucial to one’s wellbeing [13], for example, through reducing anxiety [14] and reducing stress hormones such as cortisol [10,15]. Based on the concept of noticing the good things in nature [16], this study presented a Smartphone-based wellbeing intervention designed to engage users with the good things in urban nature.

Two main theories accounting for the benefits of exposure to nature are Kaplan’s [17] Attention Restoration Theory (ART) and Ulrich’s [18] Stress Reduction Theory (SRT). ART proposes that being in and looking at nature allows the brain to recover from mental fatigue and restore attentional focus [17]. SRT proposes that nature can benefit wellbeing through its stress reducing properties [19]. For example, physiological measurements have shown that people can recover from stressful events after being exposed to nature via an increase in parasympathetic nervous system activity, thus reducing stress and arousal [20]. 

Another possible mechanism for the beneficial effects of exposure to nature is via an increase in positive emotions. Fredrickson’s [21] broaden and build theory of positive affect states that daily increases in positive emotions broaden awareness and encourage exploration, which builds skills, resources and psychological resilience over time, leading to sustained wellbeing benefits. Most studies exploring nature exposure have focused on a single dimension of positive affect [22]. However, Ulrich [23] noted two types of positive affect (positive emotional reactions to nature and wakeful relaxation) drive physiological changes related to emotion regulation. Korpela et al. [24] noted that nature provides an overlooked environment for emotional regulation and the physiological response to nature exposure has been explained with reference to models of affect regulation [25]. This study examined this by utilising a multidimensional scale of positive affect [26].

In addition to exposure to nature, the psychological construct of nature connectedness has been identified [27]. Nature connectedness, defined as an “individuals’ experiential sense of oneness with the natural world” [27], has been shown to be related to wellbeing across a number of psychological variables and validated measures (for reviews see [28]). It has importance in terms of wellbeing [29], positive affect [30], life satisfaction [27] and happiness [31]. Indeed, the wellbeing benefits of nature connectedness are estimated to be as large as established factors such as income, marital status and education [28]. The mechanisms by which nature connectedness brings about wellbeing are less well understood, but relationships to positive affect have been found [29] which suggest a link to affect regulation. Richardson and McEwan [31] found that the wellbeing benefits of nature connectedness were facilitated by emotional regulation, consistent with SRT. However, Gidlow et al. [32] found ART did not provide an explanation and Capaldi et al. [33] suggested that the wellbeing benefits of nature connectedness are not adequately described by theories developed to explain the benefits of nature exposure. In sum, nature connectedness provides both a pathway to wellbeing and can be improved in a variety of environments, including urban [16]. 

Previous studies of the benefits of natural environments to wellbeing have typically been correlational, employing spatial (Geographic Information System; GIS) analytical techniques correlating green spaces with routine health and social care data. These have shown that access to urban green spaces was associated with greater wellbeing, physical health and social contact [34,35,36,37,38,39] and lower job-related chronic stress [40]. Experience sampling methods utilising technology such as Smartphone applications [41,42], online participatory GIS [43,44,45,46] and social media [47,48] are increasingly being used to assess the relationships between urban environments and wellbeing in real time in the field and are finding that wellbeing is associated with the natural environment.

Given the benefits of nature, the mental health crisis and growing urbanity, there is a need to go beyond correlational studies and evaluate interventions designed to improve wellbeing through engaging with urban nature. Data collected from experimental studies that focus on interventions to increase people’s contact and connection with nature could be of great value to public health organisations as social prescriptions. At present, nature is an underutilised resource in public health interventions [6,49]; for this reason conservation organisations have lobbied the UK government for 1% of the public health budget to be invested in preventative nature-based solutions [50]. Our study addresses the need for evaluation of an urban nature-based intervention using an experimental design trialing a novel Smartphone-app-based intervention (called Shmapped) to improve wellbeing.

Smartphone use is high and is expected to continue growing. A recent survey showed that 81% of adults in the UK own a Smartphone [51]. Smartphones are a valuable way of reaching people, as users have been shown to unlock their phones up to 200 times per day, and to spend most of their phone time using apps [52]. This places apps in a unique position for optimising behaviour-change interventions [53]. Studies utilising Smartphone apps for data collection enable the capture of large, representative samples, have high ecological validity [54] and allow for in the moment and in the field responsiveness, although a previous study involving a Smartphone wellbeing app did have a bias toward middle-class participants [42].

Previous apps have monitored urban wellbeing, e.g., Urban Mind [41] and Mappiness [42] and found correlations between time spent in green spaces (measured through their phones Global Positioning System; GPS) and wellbeing (measured through questionnaires). However, these apps were data collection tools and did not deliver any interventions. They also found that adults only spend 7.48% of their time outdoors each day, thus, there was limited data collected on time spent in the natural environment. The current research built on this by creating a Smartphone application called Shmapped, which is a dual data collection tool and intervention which uses location-driven prompts to capture people’s wellbeing in the moment of being outdoors in publicly accessible green spaces. This was achieved through GPS positioning and geofences to locate green spaces. Although GPS and accelerometry data were recorded in this study, these will be published elsewhere.

The intervention aspect of the app is based on a positive psychology intervention that tasks people to notice ‘three good things’ daily, with consequent sustained improvements in wellbeing outcomes [55]. This awareness of positive things results in positive affect [56] which is theorised to broaden the scope of attention and improve psychological resources [57]. In previous research, the ‘three good things’ approach was adapted to notice and write about the good things in nature and resulted in increased nature connectedness which was associated with psychological wellbeing [16]. However, this research was small in scale and did not deliver significant improvements in wellbeing. Further foundation for the approach was provided by The Wildlife Trusts’ *30 Days Wild* campaign, which engaged people with everyday nature over a month and found increases in nature connectedness, positive affect and wellbeing [31]. However, this research did not recruit a control group and the participants were overwhelmingly female. The present research is larger in scale and includes a comparison group.

The Smartphone app was created to (i) monitor peoples use of green spaces, (ii) identify relationships between types of green space (i.e., woodland, wetland etc.) and wellbeing, and (iii) to act as an intervention to increase nature connectedness and wellbeing (See Figure 1 for a screenshot of the app and for a detailed description of the app and its development and feasibility testing see [58,59]). This paper focuses on the third aspect, testing the hypothesis that the wellbeing of app users will increase in both conditions because noticing the good things about one’s surroundings is not dissimilar to previous positive psychology-based interventions (Seligman et al., 2005) which have been shown to improve wellbeing. However, it was hypothesised that because of the evidence linking exposure to the natural environment and human wellbeing [4,5,6] and previous nature-based interventions improving wellbeing through noticing and connecting to nature [16], the wellbeing effects would be stronger for participants in the experimental condition who were prompted to notice nature, in contrast to a noticing built space control. It was further hypothesised that nature connectedness would increase in the experimental group and improvements in wellbeing would be related to increases in nature connectedness and positive affect. The app was also trialed as a social prescription to assess whether it would improve wellbeing in adults with common mental health difficulties. Analyses were also conducted to assess clinical significance, i.e., whether the intervention had a reliable and noticeable effect on daily life, which is more meaningful to health professionals who are monitoring whether interventions improve patient outcomes to a substantial enough level to be worth investing in.

## 2. Materials and Methods

The study randomised participants to either the green space condition or an active control (built space condition). The design was a repeated measures time-series experimental design with self-reported measures of wellbeing and nature connectedness completed in the app at three time-points: baseline, post-intervention and follow-up at one month. There was a desire to learn about the experimental treatment (i.e., to gain additional information on the green space condition and its mechanisms of action) and to maximise power, so more participants were randomised to receive it [60]. In total, 70% of participants were randomised to the green space condition; when their phones GPS recorded them as being within a green space, the app prompted them to enter one good thing they had noticed. Green spaces were identified using data provided by Sheffield City Council, which identifies all publicly accessible green and open spaces. This data was then translated into geofence data to be picked up by a Smartphone’s GPS. 30% of participants were randomised to a control condition of noticing the good things about built spaces in the same urban environment as those in the green space condition. These participants were prompted by their phone at random points during the day, with an evening reminder in order to produce an experience similar to those in the green space condition. Sending out random prompts as opposed to prompting when users were not in green spaces was necessary, as there was also no equivalent dataset identifying ‘urban or grey spaces’ held by Sheffield City Council.

This study targeted Sheffield residents who were over 18 years old and owned a Smartphone. Smartphone-based studies tend to attract middle-class adults [42]. A representative sample regarding socio-economic status was therefore targeted by trying to encourage recruitment from areas classed as higher on the 2015 English index of multiple deprivation. Moreover, part of the focus of the programme of research (Improving Wellbeing through Urban Nature—http://iwun.uk) was to look at groups with reported low exposure and connection to nature. Given that people in areas of higher deprivation have lower nature exposure (possibly due to having less access to good quality green spaces [61]), this targeted recruitment was partly to encourage residents with the greatest need to connect with nature to participate. The main strategies for promoting the Smartphone app were through social media; distributing posters and leaflets; through conservation organisations (namely the Wildlife Trusts), Council staff, large local employers, and General Practitioners (GPs). Responses indicated that social media (*n* = 408) was the most successful strategy, followed by the Wildlife Trust (*n* = 107) and posters/leaflets (*n* = 103). However, most participants found out about the study through outside these approaches as ‘other’ was selected most (*n* = 821). Participants who completed the post-intervention measures were eligible to receive a £20 voucher. Of the 1112 people who downloaded the app, 582 (54.2%) were eligible to participate (aged over 18 years and living in Sheffield as denoted by their postcode) and supplied baseline data. Of those who supplied baseline data, 322 (55.1%) completed post-intervention measures and 164 (27.4%) completed follow-up measures at 1 month. Depending on condition, built or nature, participants were asked to record a good thing about their surroundings once a day for 7 days. Those who completed the study took part between November 2017 and May 2018. 

In terms of being promoted as a social prescription, 59 participants were referred by their GP, but only nine met the reference range criteria (baseline score of ≤24) for being classed as a clinical population according to baseline scores on the Recovering Quality of Life (ReQoL) scale [62,63]. However, of the total sample supplying baseline data, 148 of participants were classed as having mental health conditions within the clinical range according to the ReQoL. Table 1 shows the participants’ demographics at each time point in the study. We also targeted as representative a population as possible in terms of ethnicity and this is shown in Table 1 in terms of representation of Black Asian Minority Ethnicity (BAME) participants.

Upon downloading the app, participants were asked to read brief information before providing consent by tapping ‘yes, I agree’ in the app. The app then asked users if they were sure they wished to consent and offered another chance to review the information sheet or decline consent. Of the 1112 participants who downloaded the app, 847 consented to participate. Users could revisit the information sheet at any time in the app. The information sheet and Privacy Impact Assessment (PIA) were also available on the study website in case participants wanted to read them before downloading the app. The study was approved by the Human Sciences Research Ethics Committee at the University of Derby and a regional research ethics committee.

After providing consent, participants were randomised to either the intervention condition (70% noticing the good things about green spaces) or the control condition (30% noticing the good things about built spaces). They were then asked to complete questionnaires within the app. Primary outcome measures included: the 10-item Recovering Quality of Life scale (ReQoL; *α* = 0.92) [62] and the single item Inclusion of Nature with Self scale (INS; *α* = 0.90) [64]. Secondary outcome measures included the 18-item Types of Positive Affect Scale (TPAS) assessing safe, relaxed and activated positive affect (*α* = 0.83 activating and relaxed positive affect, *α* = 0.73 safe positive affect) [26]; the 6-item short form Nature Relatedness scale (*α* = 0.86) [30]; and the 4-item Engagement with Natural Beauty scale (*α* = 0.87) [65]. Three items measured previous exposure to nature growing up, previous exposure to nature in the last year and whether participants had access to a garden. The ReQoL was selected as, like other measures of quality of life (QoL), it allows for health economic analysis (presented in another paper) but focuses specifically on the mental wellbeing aspect of QoL rather than just physical health. It also has an established minimum important difference, allowing for analysis of clinical significance (ReQoL Scoring, reqol.org.uk). The TPAS was selected, as unlike other unidimensional measures of positive affect, the TPAS distinguishes between calm and activated positive affect types, which may both be stimulated to different degrees by spending time in nature. The Nature Relatedness scale and INS scales are commonly used brief measures of nature connectedness and have been used in intervention studies [16,31]. Finally, the Engagement with Natural Beauty scale was used as it was previously shown to mediate the relationship between nature connectedness and wellbeing [33] and its use allowed us to look further at mechanisms of intervention effectiveness.

Given that adults only spend 7.48% of their time outside [42], green space prompts were designed to be intelligent and prompted the user whilst they were in a green space. Built space prompts were random but usually occurred around midday. If participants chose to ‘snooze’ their response, they were reminded at 8 pm, as the evening is normally a time when people start to slow down and reflect upon the day’s activities and this allowed plenty of opportunities to engage with the intervention in daylight hours. At the end of 7 days and 1 month later, participants repeated the questionnaire measures.

## 3. Results

### 3.1. Data Analysis

Data were screened for normality and found to be within acceptable ranges. Skewness ranged from −0.030 to −0.990 and kurtosis ranged from 0.085 to 1. The mean number of observations made per participant was 6.54 (SD = 3.23; range = 1–13) indicating good adherence to the app. 

A *t* test showed no significant difference in scores at baseline or the number of observations made by participants in the green and built space conditions. Analysis of the content of observations indicated good fidelity in the green space condition, with only 24 out of 367 comments (5.51%) relating to green features associated with built space (e.g., planters around buildings). Fidelity was not as good in the built condition, with 31 out of 166 comments (18.67%) exclusively about green spaces. Data were analysed using a repeated measures MANOVA (multivariate analysis of variance) with time (baseline, post, follow-up) as the within-subjects variables and condition (noticing the good things about green spaces versus built spaces) as the between-subjects variable. To assess which demographic (age, gender, ethnicity and socio-economic status) or profile of participant (low/high exposure and connection to nature at baseline) benefited the most from the intervention, demographic and baseline scores were considered as covariates. *t* tests and Chi-square were also used to assess for whom the app was least or most effective. To assess the mechanisms behind the impact of the app on wellbeing, correlations and multiple regressions were performed. The original intention was to assess whether the app could act as a social prescription to improve wellbeing in adults approaching their GP with mental health difficulties. However, only 59 patients were signposted by their GP and only nine of these met the reference range of the ReQoL to be classed as a clinical case. Therefore, we conducted a MANOVA with participants who met the reference range from the general population (*n* = 148) as a tentative examination of the effectiveness of the app as a social prescription. In particular, we assessed whether the change in wellbeing scores reached clinical significance, defined as an improvement of at least five points on the ReQoL (http://www.reqol.org.uk/p/scoring.html).

### 3.2. The Effectiveness of Noticing the Good Things in Nature

The MANOVA revealed a statistically significant difference between scores at baseline, post and follow-up (*F*(14, 111) = 4.27, *p* ≤ 0.001, η*p*^2^ = 0.350) at the multivariate level. At the univariate level, there were significant effects for all scores except the Engagement with Natural Beauty scale. There was no significant main effect of condition at the multivariate level (green vs. built space) (*F*(7, 118) = 0.964, *p* = 0.461, η*p*^2^ = 0.054). However, there was a significant time (baseline, post and follow-up) by condition (green vs. built space) interaction effect at the multivariate level (*F*(14, 111) = 2.13, *p* = 0.015, η*p*^2^ = 0.211). At the univariate level, there were no significant interaction effects. Mean scores across variables reveal improvements in all scores and can be seen in Table 2. Higher scores on variables indicate good wellbeing and nature connectedness. In sum, participants in both conditions (green and built) showed improved scores after using the app across all variables except natural beauty.

### 3.3. Noticing the Good Things in Nature as a Social Prescription

This analysis focused on participants who met the reference range for having a mental health issue according to their baseline ReQoL score (*n* = 148). The minimum important difference for scores on the ReQoL-10 measure to reach clinical significance is a 5-point increase [63]. For our sample, 78 of the 148 participants achieved a 5-point increase (*M* = 7.50, *SD* = 3.02, range = 5–19). The MANOVA showed a significant multivariate between-subjects effect [*F*(7, 116) = 16.57, *p* ≤ 0.001, η*p*^2^ = 0.500] of caseness, with significant univariate effects for the ReQoL, three types of positive affect and nature relatedness. There was a significant multivariate interaction effect (time x caseness) [*F*(14, 109) = 3.16, *p* ≤ 0.001, η*p*^2^ = 0.289], with significant univariate effects for the ReQoL (*p* ≤ 0.001). There was also a significant multivariate interaction effect (condition x caseness) [*F*(7, 166) = 2.15, *p* = 0.043, η*p*^2^ = 0.115], with significant univariate effects for the ReQoL (*p* = 0.013). These effects were explored further using a *t* test where participants were grouped according to caseness (*n*=148) or non-caseness (*n* = 452). In both the built (*t* = −2.58, *df* = 91, *p* = 0.012) and green (*t* = −5.55, *df* = 223, *p* ≤ 0.001) conditions, participants who were classed as having baseline scores on the ReQoL which indicate clinical caseness showed significantly greater improvements in the ReQoL than participants who were classed as being non-cases. In the green condition, this difference in scores exceeded the minimum important difference (change score = 5.12). In the built space condition, the difference in ReQoL scores was 3.20, thus not exceeding the minimum important difference. The implication of these results is that the improvement in scores was clinically significant only in the green space condition [63]. In summary, participants classed as having a mental health issue showed a greater improvement in scores on the ReQoL than those classed as non-cases, and participants in the green space condition showed especially greater improvements which met both statistical and clinical significance.

### 3.4. Who Benefits from Noticing the Good Things in Nature?

There was a significant multivariate between-subjects effect of time spent outside as a child [*F*(7, 117) = 5.06, *p* ≤ 0.001, η*p*^2^ = 0.233] on questionnaire scores between the green and built space conditions, with significant univariate effects across all variables except Engagement with Natural Beauty. A post-hoc *t* test comparing the green and built conditions revealed a significant effect of time (baseline, post, follow-up) in the green space condition for participants who had spent more time outdoors as a child to show a greater improvement in nature connectedness (INS) scores (*t* = 1.99, *df* = 236, *p* = 0.048). Hence, participants who spent more time outside as a child improved more on nature connectedness scores in the green condition compared with the built condition.

There was also a significant multivariate between-subjects effect of time spent outside in the last year on questionnaire scores between the green and built space conditions [*F*(7, 117) = 4.07, *p* ≤ 0.001, η*p*^2^ = 0.196] with significant univariate effects across all variables except Engagement with natural beauty. A post-hoc *t* test revealed significant effects of time (baseline, post, follow-up) in the built condition for the ReQoL (*t* = 2.67 *df* = 91, *p* = 0.009) and in the green condition for nature connectedness (NR6 t = 2.87, *df* = 232, *p* = 0.005 and INS *t* = −2.07, *df* = 236, *p* = 0.040). Participants who spent less time outdoors in the last year showed greater improvements on the ReQoL in the built condition, and those who spent less time outdoors in the last year improved more on both nature connectedness measures in the green condition.

There was a significant multivariate between-subjects effect of baseline nature connectedness score (INS) on questionnaire scores between the green and built space conditions [*F*(7, 117) = 72.99, *p* ≤ 0.001, η*p*^2^ = 0.814] and a multivariate interaction effect [*F*(14, 110) = 3.70, *p* ≤ 0.001, η*p*^2^ = 0.320] between baseline nature connectedness score and time (baseline, post, follow-up). At the univariate level, there were significant between-subject effects for all variables except the ReQoL and significant interaction effects for relaxed positive affect (*p* = 0.023) and nature connectedness (INS) (*p* ≤ 0.001). A post-hoc *t* test revealed significant effects in the green space condition with both measures of nature connectedness NR6 (*t* = −2.73, *df* = 231, *p* = 0.007) and INS ( *t*= 7.00, *df* = 236, *p* ≤ 0.001) improving more in those who had lower baseline nature connectedness (INS) scores. In summary, in the green space condition, nature connectedness scores improved most in those who started with a lower baseline in nature connectedness scores.

There were no significant effects of age, gender, ethnicity or socio-economic status (as measured by the 2015 index of multiple deprivation: IMD), (*p* > 0.05), having access to a garden, or number of observations (as a measure of engagement) on the effectiveness of the app as an intervention to improve wellbeing and nature connectedness. A Chi-square comparison of demographic data from the app with two broad classes from 2015 IMD data (high and low deprivation, measured around the median deprivation score for Sheffield) showed no significant difference (*p* > 0.05), indicating that the demographic profile of the app was no different to expected levels of socio-economic deprivation. Hence, this sample showed good representation of the population when compared with deprivation data. 

### 3.5. The Mechanisms behind the Benefits

Separate analyses were performed for the green and built space conditions. In the green space condition, correlation analysis revealed significant associations (*r* = 0.16 to 0.22) between the changes in wellbeing (ReQoL) and nature connectedness (INS) and types of positive affect (relaxed, safe & activated); these were therefore entered into a regression analysis. The analysis showed a significant model with 30% variance in the change in wellbeing explained [*F*(4, 218) 5.57, *p* ≤ 0.001]. Changes in nature connectedness (INS) (*β* = 0.21, *p* = 0.001) and relaxed positive affect (*β* = 0.16, *p* = 0.043) emerged as significant predictors of wellbeing, with safe positive affect just missing out on statistical significance (*β* = 0.15, *p* = 0.051). Activated positive affect was not a predictor. In the built space condition, none of the variables correlated with the change in wellbeing significantly; hence, regression analysis was not conducted. To sum, in the green space condition, changes to scores of nature connectedness and relaxed positive affect predicted wellbeing.

## 4. Discussion

This study assessed the effectiveness of an intervention to improve wellbeing through noticing the good things in urban nature, thus combining nature with an existing positive psychology-based intervention. There were significant increases in wellbeing and nature connectedness scores following using the app for 7 days, which were sustained at a 1-month follow-up (see Table 2 for descriptive statistics). Importantly, these differences were more pronounced in the green space condition for adults with common mental health difficulties. Hence, nature could be used to enhance an existing positive psychology-based intervention and the results here indicate that this may be a promising intervention. Furthermore, adults with mental health difficulties (according to a MANOVA focusing on participants meeting the ReQoL clinical cut-off scores) showed significantly greater improvements in the ReQoL between baseline and post than participants who were classed as being non-cases, with the difference reaching clinical significance (in addition to statistical significance) in the urban green space condition. This indicates that noticing the good things about urban nature has strong clinical potential as an intervention and social prescription for improving outcomes on wellbeing.

Noticing good things in urban nature over 7 days resulted in increased wellbeing and nature connectedness scores for participants in both the green space condition and built space condition (see Table 2 for descriptive statistics). The increase in wellbeing is consistent with evidence from positive psychology interventions such as Seligman et al. [55], because noticing the good things about ones’ surroundings is not dissimilar to previous positive psychology-based interventions. However, because of the evidence linking exposure to the natural environment and human wellbeing [4,5,6] and previous nature-based interventions improving wellbeing [31], it was hypothesised that these effects would be stronger for participants in the noticing nature condition, in contrast to noticing built space. This was supported. However, the increase in nature connectedness in those noting the good things in the built environment was unexpected. Similar to previous work with those noticing good things without a focus on nature, the level of nature connectedness at follow-up did return towards baseline, whereas it continued to rise in the green space group [16]. Analysis of the content of observations indicated some issues with fidelity in the built condition, with 18.67% of comments exclusively about green spaces. Therefore, the short-term increase in nature connectedness could be explained by noticing some aspects of nature. It is also possible that the intervention generally increased participants’ attentiveness to their surroundings.

By using an experimental design, including validated measures and making comparisons to a control group, this study provides some of the first evidence of causality that improving nature connectedness led to improving wellbeing, therefore supporting the findings from correlational research [28]. It also contributes significantly to results from other nature connectedness-based interventions which did not include a control group, such as The Wildlife Trusts’ *30 Days Wild* [31]. The evaluation of *30 Days Wild* found that engaging with nature every day improved wellbeing and nature connectedness, although unlike the current study, this was not focused within an urban environment.

### 4.1. Noticing the Good Things in Urban Nature as a Social Prescription

In terms of acting as a social prescription, the app showed promise. In both conditions, participants classed as having a mental health difficulty according to the Recovering Quality of Life scale (ReQoL), showed significantly greater improvements in the ReQoL than participants who were classed as being non-cases. In the green condition, this difference in scores exceeded the minimum important difference on the ReQoL (an improvement ≥5 points) and reached clinical significance. Maller et al. [66] advocated nature-based interventions as a basis for a socio-ecological approach to public health and a strategy in the prevention and alleviation of mental ill health, with potential application for higher risk individuals. The current work supports this approach, provides a specific methodology and extends it to a focus on nature connectedness. Regression to the mean within a sub-group is a potential issue, however, there is no reason to suspect that regression to the mean should be greater in the nature group compared to the built group. ReQoL was used to identify case or non-case, but the analysis explicitly looked at change in ReQoL. Moreover, test and retest reliability for ReQoL has been examined in both patient and general population samples by intraclass coefficients and found to be acceptable [62].

### 4.2. Who Benefits from Noticing the Good Things in Nature?

Participants who gained particular benefits from using the app included (i) participants who had spent more time outdoors as a child, who showed greater improvement in nature connectedness (INS) scores in the green space condition, (ii) participants who spent less time outdoors in the last year, who improved more on the ReQoL in the built condition and on nature connectedness in the green space condition, and (iii) those who had lower baseline nature connectedness scores improved more on nature connectedness in the green space condition. Overall, similar to *30 Days Wild* [31], this is supportive of targeting those who spend little time outside, as greater benefits of nature-based interventions were found. This also highlights the need for engagement with nature in everyday life. There is some discussion that childhood exposure to nature is important for nature connectedness as an adult [67], but there have been no longitudinal studies to evidence this, so this is an interesting finding and perhaps evidence of a ‘latent nature connectedness’. In other words, if a childhood connection with nature is reignited by using an intervention like the app, this can result in a renewed nature connectedness and subsequent wellbeing benefits.

### 4.3. The Mechanisms behind the Benefits

Building on previous literature on the wellbeing benefits of nature connectedness [28], increased nature connectedness was a predictor of increased wellbeing in participants using the app. This is consistent with previous research showing that interventions that seek to increase nature connectedness, had beneficial effects on wellbeing [16] and supports the growing importance of the psychological construct of nature connectedness as a new paradigm for wellbeing [68]. In addition, increased relaxed positive affect was a significant predictor of the improvement in wellbeing in the green space condition, which is consistent with previous literature showing that exposure to natural environments was associated with greater wellbeing than in built environments [41].

This study was the first to use a multidimensional measure of positive affect, which distinguishes low arousal/positive valence affects (such as relaxed and safe positive affects) from high arousal/positive valence affects (such as activated positive affects) as an outcome measure for a nature connectedness intervention. Low arousal positive affects, such as relaxation, have been found to uniquely predict life satisfaction, depression, wellbeing, mindfulness, anxiety, and stress beyond high arousal positive affects, such as activation [69]. The inclusion of the Types of Positive Affect Scale [26] revealed a unique finding: an intervention which increased nature connectedness and relaxed positive affect predicted increased wellbeing. This indicates a pathway which offers support for the Stress Reduction Theory [19], proposing that being in and looking at nature is restorative and reduces arousal and stress. The finding that relaxed positive affect and nature connectedness were predictors of increased wellbeing is also consistent with the affect regulation account of wellbeing [25,26,31], which states that low arousal positive affect such as relaxation and high arousal activated positive affect, such as excitement, can offer unique inputs to wellbeing through nature connectedness.

### 4.4. Limitations and Future Directions

Given the wider project requirements, timeframe and budget, engagement with the app could have been further enhanced. There was a compromise in trying to create an app that was suitable for data collection and evaluation but was at the same time engaging. A feasibility study revealed that whilst participants found the app functional, they only found it moderately engaging [58]. If engaged with more frequently, the noticing the good things in nature concept used by the app has promise as an intervention to improve wellbeing and nature connectedness. The wider mapping concept of the app also has value as a data collection tool for monitoring the quality and usage of urban green spaces so that these can be optimised to improve wellbeing. 

Numbers of participants approaching their GP with common mental health problems who were signposted through GPs were disappointing, and few of those referred were classed as clinical cases (according to baseline scores on the ReQoL). Therefore, the question about the effectiveness of the app as a social prescription was tentatively tested by taking participants from the general population who met the reference range criteria for the ReQoL. It is important to note that these individuals may not classify themselves as having a mental health issue, or be approaching their GP with a mental health issue and true testing as a social prescription will need to be a focus of future research. The study aimed to recruit 500 healthy participants and 100 adults with common mental health problems to test the feasibility of the app as a social prescription. The study exceeded the recruitment target for a healthy population (*n* = 582) but failed to recruit the target for participants presenting to their GP with common mental health problems (*n* = 59 referrals from GPs). Although GPs, IAPT and social prescription organisations were initially enthusiastic about signposting to the app, this did not translate into recruitment. On discussion with GPs, known barriers were (i) lack of time during consultation—it was felt that even handing patients a leaflet would lead to lengthy discussions, (ii) competition from other healthy living, wellbeing and physical exercise interventions, (iii) practice payments were not substantial enough to be seen as an incentive, (iv) the app is not currently an NHS approved app and was therefore seen by some as a patient-safety risk as participants may choose to write about their distress instead of writing about good things as instructed by the app. The responses during the study found no evidence to support this concern, nor did previous research where participants were asked to keep a written diary of three good things in nature [16]. When discussing social prescriptions with other organisations, a lack of signposting by GPs was a common story, which is supported by a review of social prescriptions which found that referrals from GPs were in the minority [7]. More qualitative research is required to explore the barriers and facilitators of health professionals being willing and confident to refer into social prescription interventions. It was recently recognised that social prescriptions could be a cost-effective way of reducing the burden on the NHS, with the UK Government investing £4.5 million in social prescriptions [70]. When asked in the app how participants had heard about the study, ‘other’ was the most common response. Unfortunately, ‘other’ cannot be examined further as a category, as it was the multiple-choice option within the app. This shows that the planned recruitment strategies produced fewer participants than the more unplanned, ‘viral’ approaches.

Retention rates from baseline to post-intervention (55.06%) and from post-intervention to follow-up (27.36%) were disappointing, considering that all participants completing the study at 1-month follow-up were offered a £20 voucher (see Table 1 for demographics throughout the study). This is an improvement on retention rates for an earlier 30-day version of the app in which 11.49% completed post-intervention measures [59]. Engagement with the app was compromised by the need to collect data to answer multiple research questions and required long on-boarding with questionnaires, consent and mobile phone permissions. An app simply focused on noticing the good things in nature could be much more straightforward and engaging. Finally, it is suggested that similar studies in the future should include a longer follow-up period than a month to ascertain the lasting effects of this kind of intervention.

## 5. Conclusions

Mental wellbeing and urbanisation are global issues. The study provided evidence that nature could be used to enhance an existing positive psychology-based intervention of noticing the good things in one’s surroundings to improve wellbeing. Using a novel urban social prescription implemented as a Smartphone app resulted in statistically significant improvements in wellbeing for adults in general, and statistically and clinically significant improvements in wellbeing for those classed as having a mental health difficulty. These effects were especially pronounced in the green space condition, indicating that noticing the good things about urban nature has value as a public health intervention. This study provides the first controlled experimental research evidence that a nature-based social prescription intervention can be effective in an urban environment. Providing everyday opportunities to improve wellbeing and reduce health inequalities through engaging with urban nature with a brief, portable, widely accessible and cost-effective Smartphone app intervention is of interest to public health organisations seeking solutions to mental health crises in an increasingly urbanised society [9].

## Figures and Tables

**Figure 1 ijerph-16-03373-f001:**
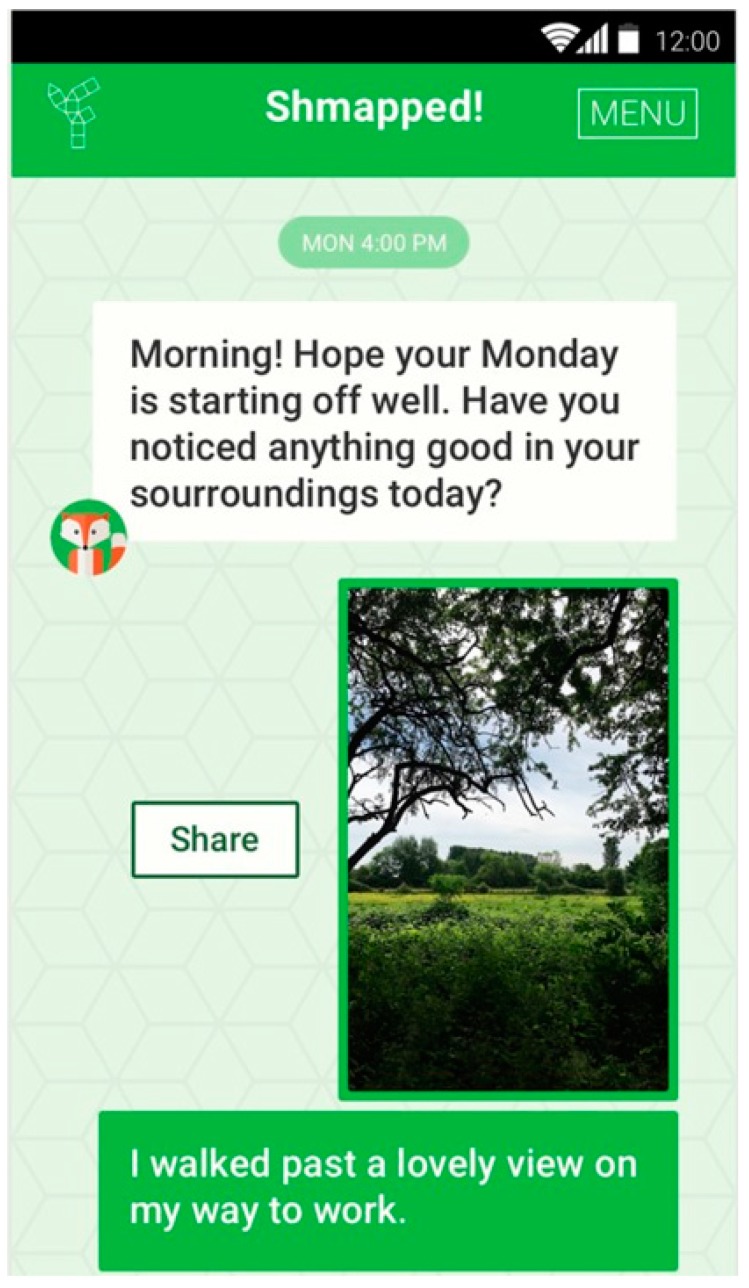
Screenshot of one of the app screens.

**Table 1 ijerph-16-03373-t001:** Participant demographics per condition at baseline, post and follow-up.

Condition		Baseline	Post	Follow-Up
Green space	*n*	414 (71.14%)	228 (70.81%)	114 (69.51%)
	*Female*	248 (59.9%)	130 (57%)	67 (58.8%)
	*Male*	164 (39.6%)	98 (43%)	47 (41.2%)
	*Average Age*	28.68 (10.43)	29.19 (10.81)	29.91 (11.17)
	*BAME*	95 (24.2%)	47 (21.5%)	18 (15.8%)
Built space	*n*	168 (28.86%)	94 (29.19%)	50 (30.49%)
	*Female*	111 (59.7%)	56 (59.6%)	28 (56%)
	*Male*	75 (40.3%)	38 (40.4%)	22 (44%)
	*Average Age*	27.75 (9.76)	27.83 (9.84)	27.52 (10.66)
	*BAME*	53 (28.5%)	18 (19.1%)	6 (12%)

**Table 2 ijerph-16-03373-t002:** Pre- and post-participation means and confidence intervals for the outcome measures.

Measure	Condition	Baseline	Post	Follow-Up
ReQol	*Green*	29.19 (28.53–29.85)	31.22 (30.39–32.05)	32.05 (30.93–33.18)
	*Built*	28.67 (27.69–29.65)	29.63 (28.21–31.06)	30.69 (28.90–32.47)
Safe	*Green*	10.41 (10.12–10.70)	10.83 (10.43–11.24)	11.47 (10.95–11.99)
	*Built*	10.65 (10.20–11.10)	11.23 (10.60–11.87)	10.77 (9.98–11.66)
Relaxed	*Green*	13.73 (13.36–14.11)	14.64 (14.15–15.12)	15.41 (14.17–16.11)
	*Built*	13.81 (13.24–14.37)	15.09 (14.17–15.61)	15.10 (14.02–16.19)
Activated	*Green*	19.16 (18.68–19.64)	19.87 (19.25–20.50)	20.63 (19.68–21.57)
	*Built*	18.88 (18.15–19.62)	20.55 (19.45–21.66)	20.65 (19.15–22.14)
Nature Relatedness (NR6)	*Green*	21.53 (21.05–22.02)	22.52 (21.88–23.17)	22.68 (21.84–23.53)
	*Built*	21.47 (20.67–22.26)	22.41 (21.20–23.62)	21.83 (20.04–23.62)
Nature connectedness (INS)	*Green*	44.23 (41.16–47.31)	49.94 (47.02–52.85)	55.40 (51.06–57.90)
	*Built*	46.77 (41.42–52.11)	52.02 (46.56–57.48)	49.85 (47.43-53.35)
Engagement with Natural Beauty	*Green*	19.30 (18.81–19.78)	19.60 (18.96–20.25)	20.19 (19.32–21.07)
	*Built*	19.36 (18.59–20.12)	19.33 (18.07–20.06)	18.71 (16.84–20.58)

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
