# Peer review of "A Smartphone App for Improving Mental Health through Connecting with Urban Nature"

_ijerph, 2019, doi:10.3390/ijerph16183373_

Round 1

Reviewer 1 Report

General comment

The intervention used in this study is unique, this will be a significant contribution in the researches on human-nature relation. However, the statistical methods used in the study were very complicated and difficult to understand the results. There is room for improvement in this regard.

Specific Comments

Consider to add some screenshots of the application.

In this study four types of analysis were performed on the data: combination of multivariate / univariate and intention to treat (ITT) / per protocol (PP) analysis. I think these four types of analysis might complicate the results of statistical tests. Authors should consider whether these four types of analysis are necessary in this study. The authors should clarify which analysis is most appropriate for analyzing the present results.

The reviewer is not familiar with ITT and PP analysis. In my understanding, ITT analysis is used to control participant dropout bias. Many participants dropped out during the intervention of this study; however, the dropout participants did not change the type of intervention. I cannot be sure that ITT analysis is necessary for this study.

The interaction between time and condition was significant only in multivariate PP analysis, and it was not significant in multivariate ITT analysis. Discussion about the differences between the two analysis results was missing.

The 147 participants were classed as clinical cases according to their ReQoL. If this ReQoL was of baseline measurement, this might be a typical "double dipping" (or circular analysis) because same data was repeatedly used for two purposes (screening and pre-post comparison). Thus the reviewer considered that the results on the participants with mental health difficulty was suspicious.

In this study, a large number of results of statistical tests were provided. The authors concluded that the intervention by using smartphone significantly improved mental health of the participants; however, it is unclear which result is based on this conclusion.

Author Response

Response to Reviewer 1

Specific Comments

Consider to add some screenshots of the application.

Authors response: We have now included a screenshot of the app in Figure 1.

In this study four types of analysis were performed on the data: combination of multivariate / univariate and intention to treat (ITT) / per protocol (PP) analysis. I think these four types of analysis might complicate the results of statistical tests. Authors should consider whether these four types of analysis are necessary in this study. The authors should clarify which analysis is most appropriate for analyzing the present results.

Authors response: ITT was an analysis we ran for completeness and to address potential discussion around drop-out rates. We found the same results from the MANOVA as the ITT so we would be happy to remove the ITT analysis for the simplicity of presenting the results.

The reviewer is not familiar with ITT and PP analysis. In my understanding, ITT analysis is used to control participant dropout bias. Many participants dropped out during the intervention of this study; however, the dropout participants did not change the type of intervention. I cannot be sure that ITT analysis is necessary for this study.

The interaction between time and condition was significant only in multivariate PP analysis, and it was not significant in multivariate ITT analysis. Discussion about the differences between the two analysis results was missing.

Authors response: We have now deleted the ITT analysis as per the reviewers comment above.

The 147 participants were classed as clinical cases according to their ReQoL. If this ReQoL was of baseline measurement, this might be a typical "double dipping" (or circular analysis) because same data was repeatedly used for two purposes (screening and pre-post comparison). Thus the reviewer considered that the results on the participants with mental health difficulty was suspicious.

Authors response: The ReQoL was not used for screening. The baseline ReQoL variable was used for post-hoc allocation to case or non-case. We have clarified this now throughout by referring to caseness as being decided by baseline scores on the ReQoL.

In this study, a large number of results of statistical tests were provided. The authors concluded that the intervention by using smartphone significantly improved mental health of the participants; however, it is unclear which result is based on this conclusion.

Authors response: We have tried to clarify which analysis in the results section the sentence refers to by only focusing on one finding at a time and clarifying in brackets which analysis the finding refers to.

Importantly, these differences were more pronounced in the green space condition for both adults with common mental health difficulties. Further, adults with mental health difficulties (according to A MANOVA focusing on participants meeting the ReQoL clinical cut-off scores) showed significantly greater improvements in the ReQoL between baseline and post than participants who were classed as being non-cases, with the difference reaching clinical significance (in addition to statistical significance) in the urban green space condition.

Reviewer 2 Report

This paper reports a randomised controlled trial designed to test the effectiveness of a positive psychology-based smartphone app on a variety of outcomes including nature connectedness, dimensions of positive affect, and quality of life. Over a week, the app prompted individuals to notice something positive about their environment: either urban nature or random depending on their assigned condition. Results suggest the app was successful in improving wellbeing and nature connectedness in both conditions, with the occasional stronger result for the urban nature condition. This study adds to the existing, primarily correlational, literature some well-needed experimental evidence regarding the effects of nature engagement and nature connectedness on wellbeing.

This paper was a very interesting read. It addresses an important issue, presents an innovative phone application and argues the importance of breaking Positive Affect up further. However, I have minor qualms about how the methods used are rationalised, and how some of the results are presented in-text, but these issues could be easily addressed with minor rewrites and more thorough explanation. Once these are explicitly justified in the text, the paper would be publishable and a good fit for this journal. Specific comments on each section are included below:

Title: Given that improving nature connectedness was your proposed mechanism of action for increasing wellbeing, and significantly increased following the trial, I think it would be helpful to include this in the title. You could also consider mentioning the fact that this is essentially a positive psychology intervention that utilises nature.

Introduction: The introduction is written well, mostly flows coherently, and pertinent background literature is referenced. Two areas would benefit from greater elaboration to help further justify the relevance and unique contribution of this study: 1) the interaction between environment type and ‘three positive things’ – an implicit premise of the study is there will be a difference in the effect of the ‘thee positive things’ based on the environment the individual is in, however the justification of this is not explicit (e.g. why will it be better in urban nature? Why would it be less effective elsewhere? Is there evidence of other contextual effects?) 2) Here and elsewhere, the population of focus is not clearly presented. Sometimes there are references to a clinical population, other times its inferred to be healthy individuals. Further clarification is need on what these population cover, who the intervention is designed to target and the relevance of looking at them (both separately and together). Please also address the following:

                Line 32-34: I am not convinced that “fastest-growing” is evidenced by the included reference. Is it necessary to state this or could it be left out?

                Line 40 (and elsewhere): The narrative around urban nature reads a little inconsistently for me. In the previous paragraph you state the benefits of urban nature but then follow up by talking about access to non-urban nature. Please be explicit in the environment type you are referring to. A better way to link this might be to reference a study demonstrating the low engagement with nature by urban populations (e.g.  Cox, Hudson, Shanahan, Fuller, & Gaston, 2017).

                Line 80: Part of your summation is that nature connectedness provides a means to engage people with urban nature, but this does not appear to be referenced.

                Line 94: I am not convinced that lobbying by NGOs constitutes an example of nature being an underutilised resource. “For example” could be switched to “For this reason”.

                Line 120-122: An opportunity has been missed to further argue the unique contribution of the paper. It would be beneficial to elaborate on this reference, explicitly critiquing it and emphasising how this study expands on those earlier findings.  

                Line 134-135: “Common mental health difficulties” is generally used to refer to mood & anxiety disorders. However, I am not sure that you assess symptoms of these at any point. Could this be changed to “poor mental wellbeing”?

Methods:

Greater detail is needed in terms of the control condition, measures used, and population selected. Control condition – the control group was a good idea, but currently the write-up implies it was poorly implemented and at risk of introducing numerous confounds. It was not clear what environments this was in and whether this information was collected (e.g. was this indoors or outdoors, as the literature would argue that this would have different impacts). The split of the sample was not also clearly rationalised, and by having such uneven numbers would actually compromise the power of the proposed analyses, thus should be justified further. Also, unlike the urban nature condition that was triggered by location, the control seemed more temporally set. Thus, might time have an influence (e.g. is urban green better because of the environment, or because that tends to happen in the morning compared to the control happening at noon or 8pm)? This could easily be checked and controlled for in the analysis if necessary. Finally, and more fundamentally, is this a true “control”? By having the same intervention (3 positive things), it is arguably a comparative intervention rather than a control. For example, causation cannot be inferred as it’s not possible to say whether these changes would have occurred over the time period anyway. This can be addressed by providing greater rationale and revising some of the terms (e.g. avoid reference to any causal effects).

Measures – Given the nature of the ReQoL I think it would make sense to include a justification of why you used this over more established measures such as the PHQ or WEMWBS. Given that your primary population was “healthy”, why did you decide not to use a measure of general wellbeing such as the WEMWBS? Alternatively, if you wanted to stratify the population by mentally healthy/mentally ill, why not use an established, symptom-based scale such as the PHQ? In this case, it is not clear to the reader exactly what the ReQoL tracks or why it was used. This also seems to have lead to difficulties in how to consistently and accurately report the results from this measure. In contrast, the TPAS has not been used widely in these types of studies, but has been justified and explained well. It was also not clear why two measures of nature connectedness were included and how they would be used. As the two look at two different definitions of connectedness (INS is more cognitive/identity whereas NR6 is more multidimensional). Please explain the justification and use of these further.

Population – A nice part of the paper and major part of its unique contribution was to the focus on accessing a more deprived population. However, the resulting sample needs to be explicitly noted, providing evidence to show that this was actually accomplished (e.g. how does this compare to the national profile?). Recruitment of a clinical population for the study appears to have either been done poorly or reported poorly. You stated that 59 participants were referred by a health professional: what kind of health professional? For what were they referred? Did they have any presenting complaints or official diagnoses upon referral? You then note that the majority of these participants did not meet your criteria for assessing poor mental health, which suggests one of two things: 1. Your attempt to recruit a clinical population was unsuccessful. Why could this be the case? Were you explicitly taking referrals with diagnoses? If not, how is this much different from taking a general population sample? OR 2. The measure you used for assessing poor mental health was not successful in discriminating the clinical sample. It would be much easier to argue against this second point had a more established or appropriate measure been used. This needs to be made clearer and also discussed at the end of the paper.

Other minor issues with the methods section include:

                Line 150-151: If prompts were given randomly, how were equal numbers of prompts across groups ensured? Did you do anything to ensure this? How did this randomisation occur (i.e. was it at a random time once each day, several times per day)? This needs more elaboration to make the methods clear and replicable.

                Line 164-165: Given that ‘other’ is your largest category (twice as large as any other), it seems pertinent to describe this category, or split it into other smaller categories.

                Line 177: “classed as clinical cases” is a little ambiguous. Clinical cases of what? Can you really call them clinical cases, or would it be more appropriate to say something like “scored in a range consistent with a clinical population”?

Results:

The results section seems to have all the key analyses reported, however the current write-up makes it difficult to follow. To help the reader through the numerous tests, it would be clearer to explain each finding, elaborating on what each one actually means. For example, it is particularly important for this paper to spend time talking through the post-hoc tests to emphasise effects of time (i.e. the evidence for the later statements that “there were significant increases… which were sustained 1 month later”) and interactions. It may also help to have a recapping statement at the end of each sub-section so the take-home findings are clear.

Referring back to the point about the clinical sample in the previous section: you analysed and reported your “social prescription” based on the population that you identified as being in poor mental health, based on your ReQoL measure. However, this was not the population that you initially recruited as a social prescription, which should presumably have been the population referred by health professionals? To me, it would have made more sense to keep these two groups separate throughout (i.e. one clinical referral group, one general population group) and analyse them separately. Were any of your clinical referral group assigned to the control condition, for example? It somewhat muddies the water when you combine groups then separate them based on different criteria. I am still not convinced that you can class your high-scoring ReQoL group as clinical, and would consider not reporting this as a separate analysis.

The following points could also be improved:

                Table 2:                 Error in upper bound of confidence interval for baseline activated affect in built condition (presume missing a 1).

Line 265: For balancing purposes, I think it is necessary to state the improvement in scores for the control condition also. As both conditions are active interventions, it is the difference in these scores that are of key importance. Does the improvement in scores for the control condition also approach clinical significance?

Line 266: “taken along with the minimum importance difference for the ReQoL” could be dropped? You reference this in the previous sentence.

Discussion:

For the most part, I feel the discussion accurately portrays, and adds interesting points to, the results. However, the role nature plays in this intervention is overstated. For me, the key takeaway is not that nature is being used as an intervention in its own right, but that nature could be used to enhance an existing positive psychology-based intervention. For example, would the intervention still have the same effect if participants were asked only to note, when prompted, the environment they were in, rather than to find a positive aspect of it? The positive psychology aspect is touched upon briefly in the main discussion, but is not mentioned at all in your conclusions.

Other things to note in the discussion section are:

                Line 315: “Adults with mental health difficulties…” by what metric is this determined? I feel this is somewhat overstated, and has been similarly overstated in other areas of the paper also. I am not sure that a 10-item self-report measure designed to track quality of life is enough to so strongly assert the presence of mental health difficulties.

                Line 321-324: Increased wellbeing and nature connectedness for the urban green condition is indeed consistent with the references listed, but is this the case for the built space condition? I think this needs more attention. Is it not an important finding that the built space condition also increased in nature connectedness? It becomes more difficult to accept your hypothesis that engagement with nature leads to higher nature connectedness and wellbeing, when the condition that was not tasked with engaging with nature also increased nature connectedness and wellbeing. Perhaps this could be explained by the intervention generally increasing attentiveness to participants’ surroundings, or by participants in the built space condition being randomly prompted while in natural spaces, but this is not addressed in further analysis or the discussion.

Author Response

Response to Reviewer 2

Title: Given that improving nature connectedness was your proposed mechanism of action for increasing wellbeing, and significantly increased following the trial, I think it would be helpful to include this in the title. You could also consider mentioning the fact that this is essentially a positive psychology intervention that utilises nature.

Authors response: We have now updated the title.

Introduction: The introduction is written well, mostly flows coherently, and pertinent background literature is referenced. Two areas would benefit from greater elaboration to help further justify the relevance and unique contribution of this study: 1) the interaction between environment type and ‘three positive things’ – an implicit premise of the study is there will be a difference in the effect of the ‘thee positive things’ based on the environment the individual is in, however the justification of this is not explicit (e.g. why will it be better in urban nature? Why would it be less effective elsewhere? Is there evidence of other contextual effects?)

Authors response: We have elaborated on the hypothesized link between environment type and three good things in the introduction as follows:

This paper focuses on the third aspect, testing the hypothesis that wellbeing of app users will increase in both conditions because noticing the good things about ones’ surroundings is not dissimilar to previous positive psychology-based interventions (Seligman et al. 2005) which have been shown to improve wellbeing. However, it was hypothesised that because of the evidence linking exposure to the natural environment and human wellbeing [4–6] and previous nature-based interventions improving wellbeing through noticing and connecting to nature [16], wellbeing effects would be stronger for participants in  the experimental condition who are prompted to notice nature, in contrast to a noticing built space control. It was further hypothesised that nature connectedness would increase in the experimental group and improvements in wellbeing would be related to increases in nature connectedness and positive affect.

2) Here and elsewhere, the population of focus is not clearly presented. Sometimes there are references to a clinical population, other times its inferred to be healthy individuals. Further clarification is need on what these population cover, who the intervention is designed to target and the relevance of looking at them (both separately and together).

Authors response: We have gone through the paper and clarified that wherever a clinical population is referred to, that this is based on baseline scores on the ReQoL meeting the reference range criteria for being a clinical case.

Please also address the following:Line 32-34: I am not convinced that “fastest-growing” is evidenced by the included reference. Is it necessary to state this or could it be left out?

Authors response: We have now edited this and removed the term ‘fasted-growing’.

Line 40 (and elsewhere): The narrative around urban nature reads a little inconsistently for me. In the previous paragraph you state the benefits of urban nature but then follow up by talking about access to non-urban nature. Please be explicit in the environment type you are referring to. A better way to link this might be to reference a study demonstrating the low engagement with nature by urban populations (e.g.  Cox, Hudson, Shanahan, Fuller, & Gaston, 2017).

Authors response: The references in para 40 [10-15) refer to urban nature rather than non-urban nature so there shouldn’t be any inconsistency. We have clarified this now in a couple of places where urban nature was referred to e.g.

In sum, nature connectedness provides both a pathway to wellbeing and can be improved in a variety of environments, including urban means to engage people with urban nature [16].

This data was then translated into geofence data by the app developers to be picked up by a smartphone’s GPS. 30% of participants were randomised to a control condition of noticing the good things about built spaces in the same urban environment as those in the green space condition. These participants were and were prompted by their phone at random point during the day, with an evening reminder in order to produce an experience similar to those in the green space condition.

Line 80: Part of your summation is that nature connectedness provides a means to engage people with urban nature, but this does not appear to be referenced.

Authors response: Nature connectedness isn’t the means to engage people with urban nature, it’s the outcome of successful engagement with nature, through, for example, noticing the good things in urban nature. This is referenced in [16], the original research which showed noticing 3 good things in nature worked, although not specifically targeting urban nature, the participants were mainly in urban/suburban environments as that’s where most people live. We have edited the text to reflect this.

Line 94: I am not convinced that lobbying by NGOs constitutes an example of nature being an underutilised resource. “For example” could be switched to “For this reason”.

Authors response: We have now edited this.

Line 120-122: An opportunity has been missed to further argue the unique contribution of the paper. It would be beneficial to elaborate on this reference, explicitly critiquing it and emphasising how this study expands on those earlier findings.  

Authors response: We have now added comments on previous research and noted how present research expands that work as follows:

However, this research was small in scale and didn’t deliver significant improvements in wellbeing. Further foundation for the approach was provided by The Wildlife Trusts’ 30 Days Wild campaign, which engaged people with everyday nature over a month and found increases in nature connectedness, positive affect and wellbeing [31]. However, this research did not involve a control group and the participants were overwhelmingly female. The present research is larger in scale and includes a comparison group.

Line 134-135: “Common mental health difficulties” is generally used to refer to mood & anxiety disorders. However, I am not sure that you assess symptoms of these at any point. Could this be changed to “poor mental wellbeing”?

Author response: The wording was taken from the ReQOL. The ReQOL measures mental health conditions, from common mental health disorders to more severe ones. A ReQoL-10 score between 0 and 24 is considered as falling within the clinical range. So that score was used as a basis for identifying common mental health difficulties. A score of 25 and above is considered as falling within the range of the general population. An increase of 5 points or more on the ReQoL-10 denotes reliable improvement. https://www.reqol.org.uk/p/overview.html

As noted above we have now clarified this throughout the paper.

For interest, ReQoL measures can be used to generate quality adjusted life years (QALYs) for use in cost effectiveness studies – economic analysis is part of the wider project.

Keetharuth, A., Brazier, J., Connell, J., Bjorner, J., Carlton, J., Taylor Buck, E., Ricketts, T., McKendrick, K., Browne, J., Croudace, T., Barkham, M. (2018). Recovering Quality of Life (ReQoL): a new generic self-reported outcome measure for use with people experiencing mental health difficulties. The British Journal of Psychiatry, 212(1)

https://www.cambridge.org/core/journals/the-british-journal-of-psychiatry/article/recovering-quality-of-life-reqol-a-new-generic-selfreported-outcome-measure-for-use-with-people-experiencing-mental-health-difficulties/3D8C7C90D326E34230E2FDBEA26AEF8D

Methods:

Greater detail is needed in terms of the control condition, measures used, and population selected.

Control condition – the control group was a good idea, but currently the write-up implies it was poorly implemented and at risk of introducing numerous confounds. It was not clear what environments this was in and whether this information was collected (e.g. was this indoors or outdoors, as the literature would argue that this would have different impacts).

Authors response: The experimental condition and control condition participants are the same population using the app. Once participants download the app, gave consent and completed baseline measures they are then randomised by the app to the green space or built space condition. Table 1 already outlines the demographics of participants in both conditions but we have also now run a t test to assess differences between conditions in terms of baseline scores, and there are no significant differences. We have added the t test finding to the results section as follows:

A t test showed no significant difference in scores at baseline or the number of observations made by participants in the green and built space conditions.

We have also elaborated on the text concerning how participants in each condition were prompted.

30% of participants were randomised to a control condition of noticing the good things about built spaces in the same urban environment as those in the green space condition. These participants were and were prompted by their phone at random points during the day, with an evening reminder in order to produce an experience similar to those in the green space condition. Sending out random prompts as opposed to prompting when users were not in green spaces was necessary as there was also no equivalent dataset identifying ‘urban or grey spaces’ held by City Council.

The split of the sample was not also clearly rationalised, and by having such uneven numbers would actually compromise the power of the proposed analyses, thus should be justified further.

Author response: Unequal group allocation is sometimes used in RCT’s to maximise power to the experimental condition so that more can be known about its effectiveness and enough participants are recruited to examine mechanisms of action (GIS location data from the experimental condition is being analysed separately to ascertain the impact of exposure to different types of natural environment – a larger dataset is required for this analysis). We have provided justification and a reference for this already but now expand slightly on this as follows:

There was a desire to learn about the experimental treatment (i.e. to gain additional information on the green space condition and its mechanisms of action) and to maximise power, so more participants were randomised to receive it [60]

Dumville, J.C.; Hahn, S.; Miles, J.N. V; Torgerson, D.J. The use of unequal randomisation ratios in clinical trials : A review. 2006, 27, 1–12.

Also, unlike the urban nature condition that was triggered by location, the control seemed more temporally set. Thus, might time have an influence (e.g. is urban green better because of the environment, or because that tends to happen in the morning compared to the control happening at noon or 8pm)? This could easily be checked and controlled for in the analysis if necessary.

Author response: We have clarified this by adding the following text:

Participants in the built-space condition were prompted by their phone at a random point during the day, with an evening reminder in order to produce an experience similar to those in the green space condition.

Finally, and more fundamentally, is this a true “control”? By having the same intervention (3 positive things), it is arguably a comparative intervention rather than a control. For example, causation cannot be inferred as it’s not possible to say whether these changes would have occurred over the time period anyway. This can be addressed by providing greater rationale and revising some of the terms (e.g. avoid reference to any causal effects).

Authors response: We have amended the text to clarify the possibility of equivalence between the experimental and control conditions. We always expected both conditions would be beneficial based on the gratitude data, but we expected the experimental condition to show stronger benefits for wellbeing:

This paper focuses on the third aspect, testing the hypothesis that the nature connectedness and wellbeing of app users will increase in both conditions because noticing the good things about ones’ surroundings is not dissimilar to previous positive psychology-based interventions (Seligman et al. 2005) which have been shown to improve wellbeing. However, it was hypothesised that because of the evidence linking exposure to the natural environment and human wellbeing [4–6] and previous nature-based interventions improving wellbeing [31] ,effects would be stronger for participants in  the experimental condition who are prompted to notice nature, in contrast to a noticing built space control.

Measures – Given the nature of the ReQoL I think it would make sense to include a justification of why you used this over more established measures such as the PHQ or WEMWBS. Given that your primary population was “healthy”, why did you decide not to use a measure of general wellbeing such as the WEMWBS? Alternatively, if you wanted to stratify the population by mentally healthy/mentally ill, why not use an established, symptom-based scale such as the PHQ? In this case, it is not clear to the reader exactly what the ReQoL tracks or why it was used. This also seems to have lead to difficulties in how to consistently and accurately report the results from this measure. In contrast, the TPAS has not been used widely in these types of studies, but has been justified and explained well.

Authors response: We have added the following text justifying our choice in using the ReQoL:

The ReQoL was selected as like other measures of quality of life (QoL) it allows for health economic analysis (presented in another paper), but focuses specifically on the mental wellbeing aspect of QoL rather than just physical health. It also has an established minimum important difference allowing for analysis of clinical significance (ReQoL Scoring, reqol.org.uk).

It was also not clear why two measures of nature connectedness were included and how they would be used. As the two look at two different definitions of connectedness (INS is more cognitive/identity whereas NR6 is more multidimensional). Please explain the justification and use of these further.

Authors response: INS and NR6 tap into different facets of nature connectedness, rather than different definitions, hence the similarity of the results when using the two measures. In a study of the range of available measures, both were found to measure the construct of Nature Connectedness (Tam, 2013). The nature of the app required a short on-boarding and the INS and NR6 are simply the briefest measures. It also gave the potential to study the difference between self/identity and affective relationships with nature. We have added the following text justifying our choice in using the remaining measures (i.e. positive affect and nature connection measures):

The TPAS was selected as unlike other unidimensional measures of positive affect, the TPAS distinguishes between calm and activated positive affect types which may both be stimulated to different degrees by spending time in nature. The Nature Relatedness scale and INS scales are commonly used brief measures of nature connectedness and have been used in intervention studies [16, 31]. Finally, the Engagement with Natural Beauty scale was used as it was previously shown to mediate the relationship between nature connectedness and wellbeing (Capaldi et al., 2017) and its use allowed us to look further at mechanisms of intervention effectiveness.

Population – A nice part of the paper and major part of its unique contribution was to the focus on accessing a more deprived population. However, the resulting sample needs to be explicitly noted, providing evidence to show that this was actually accomplished (e.g. how does this compare to the national profile?).

Authors response: We have run an analysis comparing our demographic data with census data for Sheffield and have included the following in the results section:

A Chi-square comparison of demographic data from the app with two broad classes from 2015 IMD data (high and low deprivation, measured around the median deprivation score for Sheffield) showed no significant difference (ps>.05), indicating the demographic profile of the app was no different to expected levels of socio-economic deprivation. Hence this sample showed good representation of the population when compared with deprivation data.

Recruitment of a clinical population for the study appears to have either been done poorly or reported poorly. You stated that 59 participants were referred by a health professional: what kind of health professional? For what were they referred? Did they have any presenting complaints or official diagnoses upon referral? You then note that the majority of these participants did not meet your criteria for assessing poor mental health, which suggests one of two things: 1. Your attempt to recruit a clinical population was unsuccessful. Why could this be the case? Were you explicitly taking referrals with diagnoses? If not, how is this much different from taking a general population sample? OR 2. The measure you used for assessing poor mental health was not successful in discriminating the clinical sample. It would be much easier to argue against this second point had a more established or appropriate measure been used. This needs to be made clearer and also discussed at the end of the paper.

Author response: We have changed the terminology throughout to read ‘GP’ instead of ‘health professional’. We have also clarified in the text that it was patients approaching their GP with a common mental health issue whom we asked GPs to signpost to the app:

Numbers of participants approaching their GP with common mental health problems signposted through GPs were disappointing,

Other minor issues with the methods section include:

Line 150-151: If prompts were given randomly, how were equal numbers of prompts across groups ensured? Did you do anything to ensure this? How did this randomisation occur (i.e. was it at a random time once each day, several times per day)? This needs more elaboration to make the methods clear and replicable.

Author response: We have clarified the text as follows:

Participants were prompted by their phone at a random point during the day, with an evening reminder in order to produce an experience similar to those in the green space condition.

Line 164-165: Given that ‘other’ is your largest category (twice as large as any other), it seems pertinent to describe this category, or split it into other smaller categories.

Author response: We have made note of this in the limitations section of the discussion as follows:

When asked in the app how participants had heard about the study, ‘other’ was the most common response. Unfortunately ‘Other’ cannot be examined further as a category as it was the multiple-choice option within the app. This shows that the planned recruitment strategies produced fewer participants than the more unplanned, ‘viral’ approaches.

Line 177: “classed as clinical cases” is a little ambiguous. Clinical cases of what? Can you really call them clinical cases, or would it be more appropriate to say something like “scored in a range consistent with a clinical population”?

Author response: We have added the following to the text to clarify:

 ReQOL measures mental health conditions and a “ReQoL-10 score between 0 and 24 is considered as falling within the clinical range.” https://www.reqol.org.uk/p/overview.html

Results:

The results section seems to have all the key analyses reported, however the current write-up makes it difficult to follow. To help the reader through the numerous tests, it would be clearer to explain each finding, elaborating on what each one actually means. For example, it is particularly important for this paper to spend time talking through the post-hoc tests to emphasise effects of time (i.e. the evidence for the later statements that “there were significant increases… which were sustained 1 month later”) and interactions. It may also help to have a recapping statement at the end of each sub-section so the take-home findings are clear.

Authors response: We have distinguished throughout the results what is a multivariate effect within the MANOVA and what is a univariate effect i.e. that in most cases a MANOVA was run without the need for additional post-hoc analyses owing to the lack of significant interactions at the univariate level within the MANOVA analysis. We have also added a sentence to the end of each section of the results, summing up the results in more lay-friendly terms e.g.

To sum, participants in both conditions (green and built) showed improved scores after using the app across all variables except natural beauty.

Referring back to the point about the clinical sample in the previous section: you analysed and reported your “social prescription” based on the population that you identified as being in poor mental health, based on your ReQoL measure. However, this was not the population that you initially recruited as a social prescription, which should presumably have been the population referred by health professionals? To me, it would have made more sense to keep these two groups separate throughout (i.e. one clinical referral group, one general population group) and analyse them separately. Were any of your clinical referral group assigned to the control condition, for example? It somewhat muddies the water when you combine groups then separate them based on different criteria. I am still not convinced that you can class your high-scoring ReQoL group as clinical, and would consider not reporting this as a separate analysis.

Author response: The reviewer makes a good point here and we have tried to clarify the nature of the population we have used for the social prescription hypothesis as follows and have also noted this as a limitation of the study in the discussion:

However, only 59 patients were signposted by their GP and only 9 of these met the reference range of the ReQoL to be classed as a clinical case. We therefore conducted a MANOVA with participants who met the reference range from the general population (n=148) as a tentative examination of the effectiveness of the app as a social prescription.

The question about the effectiveness of the app as a social prescription was therefore tentatively tested by taking participants from the general population who met the reference range criteria for the ReQoL. It is important to note therefore that these individuals may not classify themselves as having a mental health issue, or be approaching their GP with a mental health issue and true testing as a social prescription will need to be a focus of future research.

The following points could also be improved:Table 2:                 Error in upper bound of confidence interval for baseline activated affect in built condition (presume missing a 1).

Authors response: Thank you for pointing out this typo in Table 2 which we have now corrected.

Line 265: For balancing purposes, I think it is necessary to state the improvement in scores for the control condition also. As both conditions are active interventions, it is the difference in these scores that are of key importance. Does the improvement in scores for the control condition also approach clinical significance?

Author response: We have now added the following sentence for balancing purposes:

In the built space condition the difference in ReQoL scores was 3.20, so not exceeding the minimum important difference.

Line 266: “taken along with the minimum importance difference for the ReQoL” could be dropped? You reference this in the previous sentence.

Author response: We have now dropped this.

Discussion:

For the most part, I feel the discussion accurately portrays, and adds interesting points to, the results. However, the role nature plays in this intervention is overstated. For me, the key takeaway is not that nature is being used as an intervention in its own right, but that nature could be used to enhance an existing positive psychology-based intervention. For example, would the intervention still have the same effect if participants were asked only to note, when prompted, the environment they were in, rather than to find a positive aspect of it? The positive psychology aspect is touched upon briefly in the main discussion, but is not mentioned at all in your conclusions.

Author response: We have now made this point in the discussion and conclusions as follows:

This study assessed the effectiveness of an intervention to improve wellbeing through noticing the good things in urban nature, thus combining nature with an existing positive psychology-based intervention.

Hence, nature could be used to enhance an existing positive psychology-based intervention and result here indicate that this may be a promising intervention.

The study provided evidence that nature could be used to enhance an existing positive psychology-based intervention of noticing the good things in ones surroundings to improve wellbeing.

Other things to note in the discussion section are:

Line 315: “Adults with mental health difficulties…” by what metric is this determined? I feel this is somewhat overstated, and has been similarly overstated in other areas of the paper also. I am not sure that a 10-item self-report measure designed to track quality of life is enough to so strongly assert the presence of mental health difficulties.

Author response: As per the reviewers previous comments we have clarified this by adding the additional text (see below) and also justifying the use of the ReQoL in the methods section:

The title of the scale includes Quality of Life, but ReQOL measures mental health conditions, from common mental health disorders to more severe ones. A ReQoL-10 score between 0 and 24 is considered as falling within the clinical range. https://www.reqol.org.uk/p/overview.html

Keetharuth, A., Brazier, J., Connell, J., Bjorner, J., Carlton, J., Taylor Buck, E., Ricketts, T., McKendrick, K., Browne, J., Croudace, T., Barkham, M. (2018). Recovering Quality of Life (ReQoL): a new generic self-reported outcome measure for use with people experiencing mental health difficulties. The British Journal of Psychiatry, 212(1)

https://www.cambridge.org/core/journals/the-british-journal-of-psychiatry/article/recovering-quality-of-life-reqol-a-new-generic-selfreported-outcome-measure-for-use-with-people-experiencing-mental-health-difficulties/3D8C7C90D326E34230E2FDBEA26AEF8D

Line 321-324: Increased wellbeing and nature connectedness for the urban green condition is indeed consistent with the references listed, but is this the case for the built space condition? I think this needs more attention. Is it not an important finding that the built space condition also increased in nature connectedness? It becomes more difficult to accept your hypothesis that engagement with nature leads to higher nature connectedness and wellbeing, when the condition that was not tasked with engaging with nature also increased nature connectedness and wellbeing. Perhaps this could be explained by the intervention generally increasing attentiveness to participants’ surroundings, or by participants in the built space condition being randomly prompted while in natural spaces, but this is not addressed in further analysis or the discussion.

Authors response: We have now made this clearer in the hypotheses as follows:

This paper focuses on the third aspect, testing the hypothesis that wellbeing of app users will increase in both conditions because noticing the good things about ones’ surroundings is not dissimilar to previous positive psychology-based interventions (Seligman et al. 2005) which have been shown to improve wellbeing. However, it was hypothesised that because of the evidence linking exposure to the natural environment and human wellbeing [4–6] and previous nature-based interventions improving wellbeing through noticing and connecting to nature [16], wellbeing effects would be stronger for participants in  the experimental condition who are prompted to notice nature, in contrast to a noticing built space control. It was further hypothesised that nature connectedness would increase in the experimental group and improvements in wellbeing would be related to increases in nature connectedness and positive affect.

We have also returned to this hypothesis and these comments in the discussion as follows:

The increase in wellbeing is consistent with evidence from positive psychology interventions such as Seligman et al.’s [55], because noticing the good things about ones’ surroundings is not dissimilar to previous positive psychology-based interventions. However, because of the evidence linking exposure to the natural environment and human wellbeing [4–6] and previous nature-based interventions improving wellbeing [31], it was hypothesised that these effects would be stronger for participants in the noticing nature condition, in contrast to noticing built space. This was supported. However the increase in nature connectedness in those noting the good things in the built environment was unexpected. Previous work had found those noticing good things, without on the focus on nature, had not increased in nature connectedness [16].Analysis of the content of observations indicated some issues with fidelity in the built condition with 19% of comments exclusively about green spaces. Therefore the increase in nature connectedness could be explained by noticing some aspects of nature. It is also possible the the intervention generally increased participants’ attentiveness to their surroundings.

It is unlikely that participants in the built space condition were randomly prompted while in natural spaces as adults only spend 7.48% of their time outside [42].

Round 2

Reviewer 1 Report

I think the manuscript has been greatly improved. There is only one reviewer's concern.

The authors responded “The ReQoL was not used for screening”. I used “screening” to mean “case or non-case distinction” in the previous comment. In this study, case or non-case distinction was made by baseline ReQoL, and the baseline ReQoL was also used for a comparison with post-intervention score, thus, this is still “Double Dipping”.

Even if the intervention has no effect, a subgroup with a smaller initial score will show a higher score in the second measurement. This is called “regression to the mean”. This is a serious fallacy in the analysis. I think the results and discussion regarding ReQoL improvement in the case population should be deleted.

Even if the results and discussion regarding ReQoL in clinical cases are deleted, I think this paper is still significant.

Minor comments

Abbreviations should be defined at first mention.

Author Response

The authors responded “The ReQoL was not used for screening”. I used “screening” to mean “case or non-case distinction” in the previous comment. In this study, case or non-case distinction was made by baseline ReQoL, and the baseline ReQoL was also used for a comparison with post-intervention score, thus, this is still “Double Dipping”.

Re:Thank you for the prompt response. We’re minded not to remove the analysis as requested. Firstly, we used ReQoL to identify case or non-case but then explicitly looked at change in ReQoL. Secondly, test and retest reliability for ReQoL has been examined in both patient and general population samples by intraclass coefficients and found to be acceptable (Keetharuth et al 2018). Thirdly, there’s no reason to suspect that regression to the mean should be greater in the nature group compared to the built group. If the reviewer is correct that there may be some effect we’re not convinced that this then warrants removal of the analysis. We have though added the following to section 4.1

Even if the intervention has no effect, a subgroup with a smaller initial score will show a higher score in the second measurement. This is called “regression to the mean”. This is a serious fallacy in the analysis. I think the results and discussion regarding ReQoL improvement in the case population should be deleted.

Even if the results and discussion regarding ReQoL in clinical cases are deleted, I think this paper is still significant.

Re:Regression to the mean within a sub-group is a potential issue, however there’s no reason to suspect that regression to the mean should be greater in the nature group compared to the built sub-groups. ReQoL was used to identify case or non-case but the analysis explicitly looked at change in ReQoL. Further, test and retest reliability for ReQoL has been examined in both patient and general population samples by intraclass coefficients and found to be acceptable [62].

Minor comments

Abbreviations should be defined at first mention.

Re:Full text now added before abbreviations.